# The Role of GLP-1, GIP, MCP-1 and IGFBP-7 Biomarkers in the Development of Metabolic Disorders: A Review and Predictive Analysis in the Context of Diabetes and Obesity

**DOI:** 10.3390/biomedicines12010159

**Published:** 2024-01-11

**Authors:** Malwina Jędrysik, Krzysztof Wyszomirski, Anna Różańska-Walędziak, Emilia Grosicka-Maciąg, Maciej Walędziak, Beata Chełstowska

**Affiliations:** 1Department of Biochemistry and Laboratory Diagnostics, Faculty of Medicine, Collegium Medicum, Cardinal Stefan Wyszynski University in Warsaw, 01-938 Warsaw, Poland; m.jedrysik@uksw.edu.pl (M.J.); e.grosicka-maciag@uksw.edu.pl (E.G.-M.); b.chelstowska@uksw.edu.pl (B.C.); 2Department of Human Physiology and Pathophysiology, Faculty of Medicine, Collegium Medicum, Cardinal Stefan Wyszynski University in Warsaw, 01-938 Warsaw, Poland; k.wyszomirski@uksw.edu.pl; 3Department of General, Oncological, Metabolic and Thoracic Surgery, Military Institute of Medicine—National Research Institute, Szaserów 128 St., 04-141 Warsaw, Poland; maciej.waledziak@gmail.com

**Keywords:** diabetes, obesity, GLP-1, GIP, MCP-1, IGFBP-7, biomarkers

## Abstract

Metabolic illnesses, including obesity and type 2 diabetes, have become worldwide epidemics that have an effect on public health. Clinical investigations and further exploration of these mechanisms could lead to innovative, effective, and personalized treatment strategies for individuals. It is important to screen biomarkers in previous studies to discover what is missing. Glucagon-like peptide-1′s role in insulin secretion and glucose control highlights its diagnostic and therapeutic potential. Glucose-dependent insulinotropic peptide’s influence on postprandial satiety and weight management signifies its importance in understanding metabolic processes. Monocyte chemoattractant protein-1′s involvement in inflammation and insulin resistance underlines its value as a diagnostic marker. Insulin-like growth factor-binding protein-7’s association with insulin sensitivity and kidney function presents it as a potential target for these diseases’ management. In validating these biomarkers, it will be easier to reflect pathophysiological processes, and clinicians will be able to better assess disease severity, monitor disease progression, and tailor treatment strategies. The purpose of the study was to elucidate the significance of identifying novel biomarkers for type 2 diabetes mellitus and obesity, which can revolutionize early detection, risk assessment, and personalized treatment strategies. Standard literature searches of PubMed (MEDLINE), EMBASE, and Cochrane Library were conducted in the year 2023 to identify both original RCTs and recent systematic reviews that have explored the importance of identifying novel biomarkers for T2D and obesity. This search produced 1964 results, and then was reduced to randomized controlled trial and systematic reviews, producing 145 results and 44 results, respectively. Researchers have discovered potential associations between type 2 diabetes mellitus and obesity and the biomarkers glucagon-like peptide-1, glucose-dependent insulinotropic peptide, monocyte chemoattractant protein-1, and insulin-like growth factor-binding protein-7. Understanding the role of those biomarkers in disease pathogenesis offers hope for improving diagnostics, personalized treatment, and prevention strategies.

## 1. Introduction

Type 2 diabetes (T2D) and obesity are common civilian diseases and two interconnected metabolic disorders that have reached epidemic proportions worldwide. Roughly 90% of all instances of diabetes are type 2, making it the most common form. Insulin resistance is a characteristic that causes the body to react to insulin less effectively. Rising blood glucose levels prevent insulin from working properly, which leads to the release of more insulin. This may cause the pancreas to gradually wear out in some type 2 diabetic people, which will lead to the body producing less and less insulin and even higher blood sugar levels (hyperglycemia) [1,2]. Additionally, diabetes has a substantial burden on individuals and healthcare systems due to its high prevalence, economic costs, and increased risk of complications, and comorbidities. For instance, cardiovascular diseases, such as coronary artery disease and stroke, are complications of diabetes, leading to significant morbidity and mortality [3]. Chronic kidney disease, another complication of diabetes, contributes to end-stage renal disease and requires costly treatments like dialysis or transplantation [4,5]. Biomarker identification is crucial as it enables early detection, accurate diagnosis, and monitoring of disease progression and treatment response. Scientific examples include circulating microRNAs and glycated hemoglobin (HbA1c) as diagnostic and prognostic markers [6,7,8]. This method has some limitations, such as individual variability in the relationship between average blood glucose levels and HbA1c. Some people may have higher or lower HbA1c levels for a given average glucose concentration due to factors such as age, race, and genetic differences. Relying solely on HbA1c without considering other indicators may lead to misinterpretation. However, by identifying new reliable biomarkers, healthcare professionals can enhance risk assessment, tailor treatment strategies, and ultimately mitigate the burden of metabolic disorders on public health.

Obesity, on the other hand, is a condition characterized by excessive accumulation of body fat understood as the accumulation of triacylglycerols in adipocytes, which results in an increased number of those cells and thus higher body weight in patients [1,2]. Obesity has also become a global epidemic, presenting a significant public health challenge in recent years. Defined as an excess accumulation of body fat, obesity is a complex multifactorial disorder influenced by genetic, environmental, and behavioral factors [9,10]. Understanding the etiology and consequences of and potential interventions for obesity is of the utmost importance in addressing this pressing health issue. The pathogenesis of obesity involves a dysregulation of energy balance, characterized by an imbalance between energy intake and expenditure [11]. Adipose tissue is now understood to be an active endocrine organ that secretes a variety of bioactive chemicals known as adipokines. It was previously thought to be a passive energy (triacylglycerides) storage organ, but this view is changing. These adipokines are essential for maintaining insulin sensitivity, energy homeostasis, inflammation, and appetite regulation [12]. To effectively combat obesity and its associated health complications, it is essential to identify reliable biomarkers that can aid in early detection, risk assessment, and monitoring of treatment outcomes.

The intricate interplay of metabolic illnesses is shown by the link between diabetes and obesity. Obesity and diabetes, especially type 2 diabetes, are commonly linked in both directions, creating a synergistic hub of metabolic dysregulation. Insulin resistance, a condition wherein cells in the body lose their sensitivity to insulin and blood glucose levels rise, is a feature shared by both disorders [1,3]. Obesity, especially visceral adiposity, is a significant risk factor for insulin resistance and type 2 diabetes. The similarities lie in the underlying mechanisms of inflammation and metabolic dysfunction, wherein adipose tissue plays a central role [11,12]. Despite these shared aspects, differences exist in the temporal progression and manifestation of these disorders. Biomarkers are measurable indicators that can reflect the presence, progression, or response to treatment of a disease. In the context of T2D and obesity, biomarkers hold immense potential for improving early detection, risk stratification, and personalized treatment approaches [1]. Firstly, T2D is often diagnosed in the late stages, after significant damage has occurred, leading to increased morbidity and mortality rates. Additionally, obesity, a major risk factor for T2D, is a complex condition with diverse underlying mechanisms and phenotypes. Biomarkers specific to obesity can aid in understanding its heterogeneity, allowing for targeted interventions and personalized approaches [1,3]. Secondly, biomarkers play a crucial role in monitoring disease progression and assessing treatment response. Traditional clinical parameters, such as blood glucose levels or body mass index, have limitations in capturing the multifaceted aspects of T2D and obesity [2]. Lastly, biomarkers have the potential to facilitate the development of innovative treatment therapies. Moreover, they can help identify possible treatment targets by clarifying the biological processes and mechanisms involved in T2D and obesity [12]. Notably, markers such as glucagon-like peptide-1 (GLP-1), glucose-dependent insulinotropic peptide (GIP), monocyte chemoattractant protein-1 (MCP-1), and insulin-like growth factor-binding protein 7 (IGFBP-7), while implicated in both diabetes and obesity, may exhibit distinct patterns and quantitative variations at different stages. Gaining an understanding of the complex roles played by these indicators in the complex landscapes of obesity and diabetes can help identify commonalities and differences between these related metabolic disorders [1,3,7,11,12,13].

This paper will review relevant scientific publications from the past few years. By highlighting recent discoveries and ongoing research efforts, we aim to underscore the importance of identifying novel biomarkers and their potential implications for improving the diagnosis, management, and treatment of T2D and obesity.

## 2. Materials and Methods

The PubMed (MEDLINE), EMBASE, and Cochrane Library databases were searched for original articles, randomized controlled trials (RCTs), and recent systematic reviews that had explored the importance of identifying novel biomarkers for type 2 diabetes (T2D) and obesity, and that had been published between 2003 and 2023. The research was performed by 4 independent researchers under the supervision of the senior researcher. The following key words were used in selecting original publications regarding the role of biomarkers in T2D: “type 2 diabetes” and “biomarkers”. This search returned 1964 results, and then was reduced to randomized controlled trials and systematic reviews, with 145 results and 44 results, respectively. Regarding the role of biomarkers in obesity, a similar search was performed. The following key words were used in selecting original publications regarding the role of biomarkers in obesity: “obesity” and “biomarkers”. The second search returned 23,617 results, and then the randomized controlled trial and systematic reviews filters were used, returning 1608 and 427 results, respectively. The search was supplemented with a manual search of reference lists. When using more filters and applying particular names of biomarkers that are the aim of this publication’s consideration, fewer results were obtained, which allowed for the selection of relevant publications.

## 3. Results

The biomarkers identified as potential factors associated with the development of metabolic disorders such as type 2 diabetes and obesity are described below.

### 3.1. GLP-1

GLP-1 is a biomarker that has been extensively studied and identified as a potential factor associated with the development of metabolic disorders, including T2D and obesity. GLP-1 is an incretin hormone secreted by the intestinal L-cells in response to food intake. It plays a crucial role in regulating glucose homeostasis and insulin secretion.

#### 3.1.1. GLP-1 and Its Impaired Function in T2D Patients

In individuals with type 2 diabetes, there is often a deficiency or impaired function of GLP-1, leading to reduced insulin secretion and impaired glucose control (Figure 1) [13,14]. Numerous scientific studies have investigated the role of GLP-1 in metabolic disorders. For example, a study by Drucker et al. demonstrated that GLP-1 receptor agonists such as exenatide and liraglutide improve glycemic control and promote weight loss in patients with type 2 diabetes [15]. These agonists mimic the action of endogenous GLP-1 and enhance insulin secretion, suppress glucagon release, delay gastric emptying, and promote satiety [16,17]. Another study by Nauck et al. showed that GLP-1 receptor agonists not only improve glycemic control but also have beneficial effects on cardiovascular outcomes, reducing the risk of major adverse cardiovascular events [18].

#### 3.1.2. Dysregulated GLP-1 Secretion in Obese Patients

Moreover, GLP-1 has also been found to play a role in obesity. In obese individuals, GLP-1 secretion is often dysregulated, leading to reduced GLP-1 levels and impaired satiety signaling [19]. Research by Ard et al. demonstrated that administration of GLP-1 to obese individuals resulted in reduced food intake and increased feelings of fullness [20].

#### 3.1.3. GLP-1 as a Therapeutic Hope

Additionally, GLP-1-based therapies have shown promising results in clinical trials. The LEADER trial, conducted by Marso et al. demonstrated that the GLP-1 receptor agonist liraglutide reduced the risk of major adverse cardiovascular events and cardiovascular mortality in patients with type 2 diabetes [21]. Another study by Davies et al. showed that exenatide, a GLP-1 receptor agonist, led to significant improvements in glycemic control and weight reduction in obese individuals with type 2 diabetes [22]. Further, they emphasize the potential of GLP-1 as an important biomarker for early diagnostics and therapeutic intervention in metabolic disorders [23,24,25,26].

### 3.2. GIP

GIP, also known as gastric inhibitory polypeptide, is a biomarker associated with metabolic disorders, including type 2 diabetes and obesity. GIP, secreted by intestinal K-cells in response to nutrients like glucose and fatty acids, stimulates insulin release and adipose tissue energy storage. GIP acts as a bifunctional stabilizer of blood glucose and stimulates glucagon secretion at lower glucose concentrations [27]. Some effects of GIP differ from those of GLP-1: under conditions of insulin resistance (IR), it promotes lipid deposition in mice subcutaneous fat cells; it does not affect gastric emptying; and its effect on blood glucose is impaired in chronic hyperglycemia [1,28,29]. Pathological glucose intolerance results in an aberrant incretin effect.

#### 3.2.1. GIP’s Role in T2D Dysregulations

When type 2 diabetes patients are compared to healthy individuals who respond to oral glucose in a dose-dependent manner, the former group either has reduced GIP levels or has developed beta-cell resistance to GIP. Diabetics’ glucose intolerance is caused by a diminished influence of incretins, since incretins provide 70% of the insulin response following meals [30,31,32].

#### 3.2.2. Lipid Metabolism and GIP Correlation

Furthermore, lipid metabolism and the development of obesity depend on GIP. Obesity is associated with increased GIP levels and K-cell hyperplasia because fat is a potent stimulant of GIP release. Anabolic hormone GIP stimulates lipogenesis while preventing lipolysis (Figure 2) [2,30,31,32].

#### 3.2.3. GIP as the Main Object of Interest in Clinical Research

Clinical studies have examined GIP modulation. Christensen et al. found improved glycemic control and insulin sensitivity in type 2 diabetes patients with GIP receptor antagonism [33]. Aulinger et al. reported reduced weight and improved glucose tolerance in overweight individuals with GIP receptor antagonism [34]. Research by Otten et al. (2019) suggests that weight loss elevates postprandial GLP-1 levels, aiding weight maintenance. It also hints at the Paleolithic diet’s potential to enhance weight loss and postprandial GIP levels [35]. Stentz et al. compared high-protein (HP) and high-carbohydrate (HC) diets in prediabetes. Both led to weight loss, improved glucose metabolism, insulin sensitivity, and more [36]. However, the HP diet exhibited a higher prediabetes remission rate, possibly due to increased GLP-1 and GIP levels, enhancing insulin secretion and glucose handling.

### 3.3. MCP-1

MCP-1, also known as CCL2, is a biomarker linked to metabolic disorders like type 2 diabetes and obesity. MCP-1 is a chemokine that recruits immune cells to sites of inflammation. Studies, such as Nbdspqibhf et al. have found elevated MCP-1 in the adipose tissue of obese individuals, contributing to chronic inflammation [37].

#### MCP-1 and Its Correlation with Insulin Resistance

Kamei et al. showed that MCP-1 induces insulin resistance in mice, while blocking MCP-1 signaling improves insulin sensitivity (Figure 3) [38]. Clinical studies like those of Herder et al. and Chavey et al. associated elevated MCP-1 with an increased risk of type 2 diabetes and insulin resistance in obese individuals [39,40]. Tamura et al. demonstrated that MCP-1 receptor antagonist treatment in obese mice improved glucose tolerance and reduced inflammation [41]. Jamialahmadi et al. conducted a meta-analysis revealing a decrease in MCP-1 after bariatric surgery, suggesting an anti-inflammatory effect [42]. This reduction may contribute to improved cardiometabolic outcomes in obesity beyond weight loss. Furthermore, Ngcobo et al. (2022) highlighted MCP-1’s role in diagnosing type 2 diabetes (T2D) and its complications. Therapies like thiazolidinediones and statins have shown efficacy in reducing MCP-1 levels and inflammation in T2D patients [43]. Emerging therapies, including micronutrients and dietary supplements, also hold promise in modulating inflammatory markers and vascular function in T2D patients.

### 3.4. IGFBP7

IGFBP-7 is a biomarker linked to metabolic disorders like type 2 diabetes and obesity. It regulates insulin-like growth factors (IGFs) and affects insulin sensitivity, adipocyte function, and glucose metabolism (Figure 4) [44,45,46,47]. Watanabe et al. found that TGF-β1 enhances IGFBP7 expression via the Smad2/4 pathway in renal tubular cells [48].

#### The Urine-Based Marker IGFBP7 with Its Association with Insulin Resistance and Metabolic Syndrome

IGFBP7 is involved in epithelial–mesenchymal transition (EMT) in these cells and is associated with tubular injury and fibrosis in diabetic kidneys [48]. Thiele et al. explored empagliflozin’s effects in acute decompensated heart failure (HF) patients [49]. While it did not impact cardiac parameters, empagliflozin significantly reduced markers of acute kidney injury (AKI), specifically TIMP-2 and IGFBP7, indicating protection against tubular kidney damage. Januzzi et al. investigated IGFBP7 in heart failure with preserved ejection fraction (HFpEF). Higher IGFBP7 levels were associated with left atrial dilation and abnormal diastolic filling. Sacubitril/valsartan treatment reduced IGFBP7 levels, suggesting its involvement in HFpEF pathophysiology, particularly diastolic dysfunction and left atrial enlargement [50].

### 3.5. New Treatment Strategies: Are They Connected to Identified Biomarkers?

#### 3.5.1. Bifunctional Agonists Targeting GLP-1 and Glucagon Receptors: A Dual Approach to Managing Glucose Intolerance and Obesity

In the context of developing modern therapies, the journey commenced with the creation of bifunctional agonists, combining features of GLP-1 and glucagon, to address glucose intolerance and obesity. GLP-1 exerts antidiabetic effects, while glucagon’s hyperglycemic effects may seem counterintuitive [16,17,51]. Glucagon also increases energy expenditure and decreases food intake by inhibiting lipid synthesis and stimulating lipolysis, which causes browning of adipose tissue [51]. In rats fed a high-fat diet, a logically constructed GLP-1 and glucagon receptor bifunctional agonist successfully restored glucose tolerance and reduced obesity.

#### 3.5.2. Innovative Approaches in Diabetes and Obesity Therapeutics

Subsequently, numerous bifunctional and even trifunctional agonist combinations, some currently under clinical evaluation for type 2 diabetes and obesity, have been developed [51,52]. Based on phase III trial outcomes, tirzepatide, a bifunctional glucose-dependent insulinotropic polypeptide receptor and GLP-1 receptor agonist, gained FDA approval for treating adults with type 2 diabetes [53,54,55]. Interestingly, this compound induces clinically significant weight loss in type 2 diabetes patients. Moreover, investigations combine GLP-1 receptor agonists with nuclear hormones like estrogen or thyroid hormone to limit their impact on GLP-1R-expressing cells [56]. Patients with pathogenic variants in the leptin-melanocortin pathway or MC4R mutations who are ineligible for setmelanotide treatment may benefit from these advances in unimolecular polypharmacology. Further, the focus shifts to incretin-based therapies.

#### 3.5.3. Advancements in Diabetes and Obesity Therapeutics: Polyagonists and Structurally Selective Agonists Pioneering New Treatment Modalities

Studies optimizing structurally selective GLP-1R and GIPR agonists explore the concept that mammalian energy balance regulation involves multiple hormones. A pivotal breakthrough in this direction was the discovery of polyagonists that simultaneously interact with GLP-1, GIP, and/or glucagon receptors. Several drug candidates like GLP-1/glucagon dual agonists, GIP/GLP1 dual agonists, GIP/GLP1/glucagon tri-agonists, GIPR agonists, and GLP1R agonists have progressed to clinical trials, a comprehensive list of which can be found at Home—ClinicalTrials.gov [56,57,58,59]. Clinical investigations and further exploration of these mechanisms could lead to innovative, effective, and personalized treatment strategies for individuals affected by these conditions.

#### 3.5.4. Potential Diagnostic, Prognostic, and Monitoring Significance of These Biomarkers in the Context of Diabetes and Obesity

The biomarkers GLP-1, GIP, MCP-1, and IGFBP7 hold promise in improving diagnostics for diabetes and obesity. Impaired GLP-1 signaling has been linked to reduced insulin secretion, impaired glucose control, and increased cardiovascular risk. GLP-1 receptor agonists, such as semaglutide and liraglutide, have shown efficacy in lowering HbA1c levels and promoting weight loss in patients with type 2 diabetes (T2D) [13,14]. These agents have demonstrated superior efficacy in achieving glycemic targets compared to traditional antidiabetic medications. Measuring GLP-1 levels could provide valuable diagnostic information, aiding in disease severity assessment, treatment response prediction, and personalized therapeutic strategies [47,59,60]. GIP, another biomarker, has been associated with postprandial satiety and weight maintenance. Changes in GIP levels due to weight loss and diet composition suggest its potential role in regulating glucose metabolism and insulin sensitivity [61]. MCP-1, known for its role in atherogenesis and inflammation, has emerged as a diagnostic marker for obesity. Bariatric surgery has been found to reduce MCP-1 levels, contributing to improved cardiometabolic outcomes. Therapeutic agents targeting MCP-1 and its downstream inflammatory pathways, such as thiazolidinediones and statins, have shown efficacy in reducing monocyte activation and inflammation [62]. IGFBP7 has been implicated in diabetic nephropathy, with elevated levels observed in the urine of affected patients. Empagliflozin, an SGLT2 inhibitor, has shown renoprotective effects by reducing markers of acute kidney injury (AKI), including IGFBP7. In heart failure, IGFBP7 has been associated with abnormalities in diastolic filling and left atrial dilation [63,64]. The measurement of these biomarkers can improve diagnostics, guide treatment decisions, and enhance patient care in diabetes and obesity, which is presented in Table 1. Further research is needed to explore their mechanisms of action and long-term effects.

To highlight the importance of the proposed biomarkers in the diagnosis of diabetes and obesity; their prevalence in both conditions; their changes in the early, middle, and late stages of these two diseases; and the differences in the quantitative level of the same marker at different stages of the disease, Table 2 was prepared [64].

## 4. Discussion

Type 2 diabetes (T2D) and obesity are two very common metabolic disorders that have reached epidemic proportions worldwide, and they are classified as civilian diseases [1,2,3,4,5]. T2D is characterized by chronic hyperglycemia resulting from a combination of insulin resistance and impaired insulin secretion [1]. Obesity, on the other hand, is a condition characterized by excessive accumulation of body fat. The rise in the prevalence of both T2D and obesity poses significant health challenges and highlights the need for a deeper understanding of their underlying mechanisms [10]. Biomarkers identified as potential factors associated with the development of metabolic disorders such as type 2 diabetes and obesity, namely GLP-1, GIP, MCP-1, and IGFBP-7, have been the subject of extensive scientific research. These biomarkers have shown promise in elucidating the pathophysiological processes involved in the development and progression of metabolic disorders, offering insights into potential diagnostic and therapeutic avenues.

Taking all of this together, biomarkers, including GLP-1, GIP, MCP-1, and IGFBP-7, are valuable in the field of metabolic disorders such as type 2 diabetes (T2D) and obesity, and they can also be used as predictive models, because they provide specific information about the underlying mechanisms and pathophysiological processes of these conditions. It must be highlighted that they differ from current diagnostics, therapeutics, and clinical implications, which is pointed out below.

### 4.1. General Information

GLP-1 provides insights into insulin secretion and glucose control. It can reveal whether an individual is experiencing reduced insulin secretion, a common feature of T2D [13,14,19]. GIP levels are associated with postprandial satiety and weight management, offering information about appetite regulation and metabolism [30,31,60]. MCP-1 is linked to inflammation and insulin resistance, giving information about the presence of chronic low-grade inflammation, which is a common factor in obesity and T2D [37,38,39]. IGFBP-7 is associated with insulin sensitivity and kidney function, offering insights into metabolic status and renal health [48,49,50].

### 4.2. Innovations in Diagnostics and Therapeutics

#### 4.2.1. Specificity

Biomarkers GLP-1, GIP, MCP-1, and IGFBP-7 provide more specific information about metabolic processes than traditional diagnostic parameters like blood glucose levels or body mass index (BMI). They focus on key molecular and physiological aspects of the conditions [1,58,59].

#### 4.2.2. Personalization

Biomarkers GLP-1, GIP, MCP-1, and IGFBP-7 are enabling a personalized approach to diagnostics and treatment. Current diagnostics and therapeutics are often one-size-fits-all, whereas biomarkers allow healthcare professionals to tailor interventions based on individual profiles [1,58,59].

#### 4.2.3. Insight into Mechanisms

Unlike traditional diagnostic markers that indicate the presence of the disease, these biomarkers offer insights into the underlying mechanisms. This knowledge can help identify therapeutic targets and guide the development of more effective treatments [1,59].

#### 4.2.4. Monitoring and Progress Assessment

Biomarkers facilitate the monitoring of disease progression and treatment response. They offer a more comprehensive view of patient status compared to single-point diagnostic measurements [1,55].

### 4.3. Clinical Implications

#### 4.3.1. Early Detection

These biomarkers contribute to early detection, enabling interventions at an earlier stage of the diseases. Current diagnostics may often identify conditions when they are more advanced [59,60].

#### 4.3.2. Tailored Treatment

Biomarkers guide the selection of specific treatments, such as GLP-1 receptor agonists or GIP modulators, which are tailored to an individual’s needs. Traditional treatments are not as precisely targeted [59,60,61].

#### 4.3.3. Preventive Measures

By identifying individuals at risk based on biomarker profiles, healthcare professionals can implement preventive strategies to mitigate the development of T2D and obesity [55].

#### 4.3.4. Innovative Therapies

Understanding mechanisms through biomarkers can lead to the development of innovative therapies. Current therapies are often based on established approaches, and may not be as novel or targeted [60,61,62,63,64].

Using a panel of multiple biomarkers in combination for the diagnosis and treatment of metabolic disorders like type 2 diabetes (T2D) and obesity can offer several advantages over relying on a single marker, such as comprehensive assessment; metabolic disorders are often complex and multifactorial conditions. Using a panel of biomarkers allows for a more comprehensive assessment of the various underlying mechanisms contributing to the disease. Different biomarkers can capture different aspects of the disorder, providing a more holistic view of the patient’s metabolic status [44,63], and increased sensitivity, as some metabolic disorders may not be adequately captured by a single biomarker. By combining multiple markers, healthcare professionals can increase the sensitivity of their diagnostic approach. This is particularly important in the early detection of these conditions, when individual markers may not show significant abnormalities [54,55,56]. Using a panel of biomarkers can also help in risk stratification. Different individuals may have different metabolic profiles and risk factors. By analyzing multiple biomarkers, healthcare professionals can categorize patients into more specific risk groups, allowing for targeted interventions [8,15]. Just as with diagnosis, a panel of biomarkers can assist in tailoring treatment strategies. Different biomarkers can guide the selection of specific medications or therapeutic approaches based on the patient’s individual metabolic profile. This personalized approach is more likely to yield effective results [8]. A panel of biomarkers can also be valuable in monitoring disease progression and treatment response. Changes in multiple biomarkers over time can provide insights into the effectiveness of the chosen treatment and help in adapting interventions as needed [30,56]. Using multiple biomarkers can help reduce the risk of false positives and false negatives in diagnosis. When relying on a single marker, there is a higher chance of misclassification. Combining markers can improve the accuracy of diagnosis and reduce the likelihood of making incorrect clinical decisions [1,55]. Discussing a panel of biomarkers with patients can enhance their understanding of the complexity of their condition. Patients may be more motivated to engage in their treatment when they see a comprehensive overview of their metabolic health.

In conclusion, using a panel of multiple biomarkers offers a more informative and comprehensive approach to the diagnosis and treatment of metabolic disorders. It enhances sensitivity, specificity, and individualization in patient care, ultimately leading to more accurate diagnosis and better treatment outcomes. It is a valuable strategy within the increasingly personalized and data-driven approach to healthcare.

## 5. Conclusions

The identification and exploration of biomarkers such as GLP-1, GIP, MCP-1, and IGFBP-7 have provided valuable insights into the complex pathophysiology of metabolic disorders, including type 2 diabetes and obesity [1,2,3,4,5,6,7,8,9]. These biomarkers have shown promise in aiding early diagnosis, risk stratification, and personalized therapeutic approaches. Integrating these biomarkers into clinical practice has the potential to improve outcomes, facilitate targeted interventions, and ultimately mitigate the burden of metabolic disorders on individuals and healthcare systems [1,2,3,4,5,6,7,8,9,35]. However, further research is required to validate and refine the utility of these biomarkers, establish standardized measurement methods which will be cheap and widely available, and unravel their precise molecular mechanisms in the context of metabolic disorders.

In this regard, we propose that the above-described biomarkers be incorporated into the routine tests recommended in diabetes and obesity in order to improve the quality of diagnosis, and to diagnose patients in the early stages of the development metabolic diseases so that they can be halted; in doing so, we may reduce the risk of diseases that are complications of diabetes or obesity.

## Figures and Tables

**Figure 1 biomedicines-12-00159-f001:**
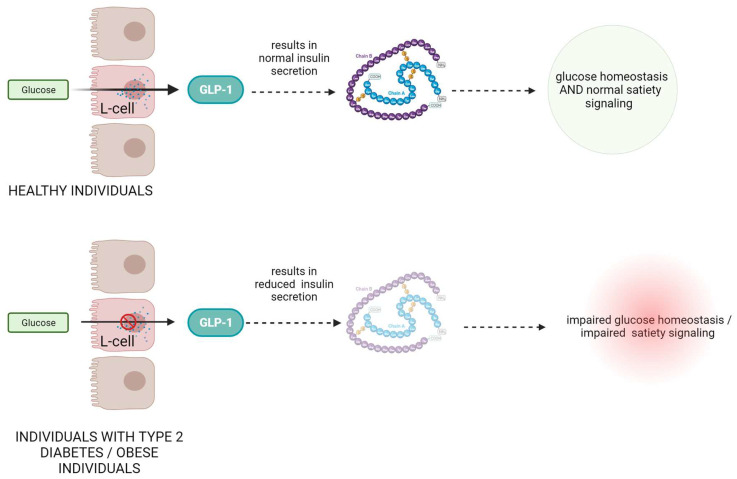
The role and mechanism of action of GLP-1 protein in healthy subjects and in patients with type II diabetes and obesity.

**Figure 2 biomedicines-12-00159-f002:**
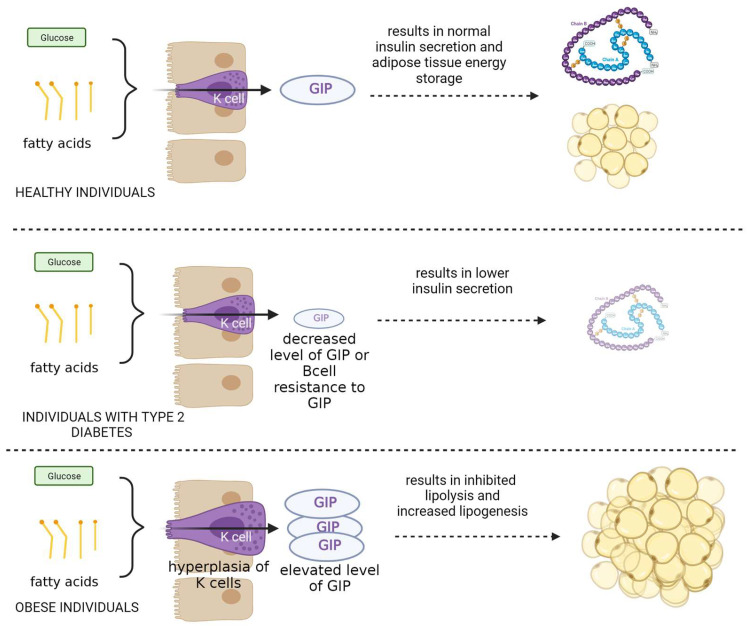
The role and mechanism of action of GIP protein in healthy subjects and in patients with type II diabetes and obesity.

**Figure 3 biomedicines-12-00159-f003:**
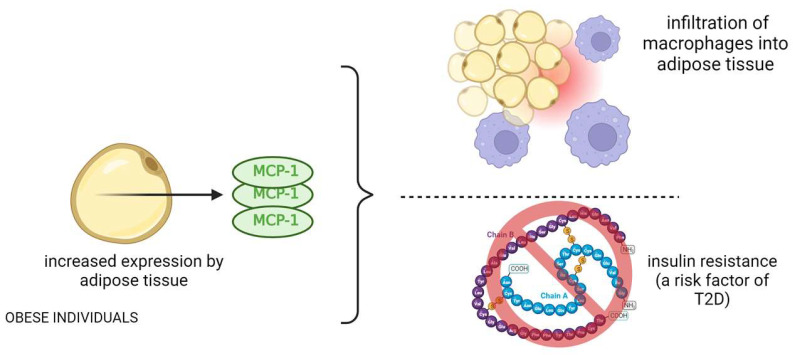
The role and mechanism of action of MCP-1 protein in obese patients and its effects related to type 2 diabetes.

**Figure 4 biomedicines-12-00159-f004:**
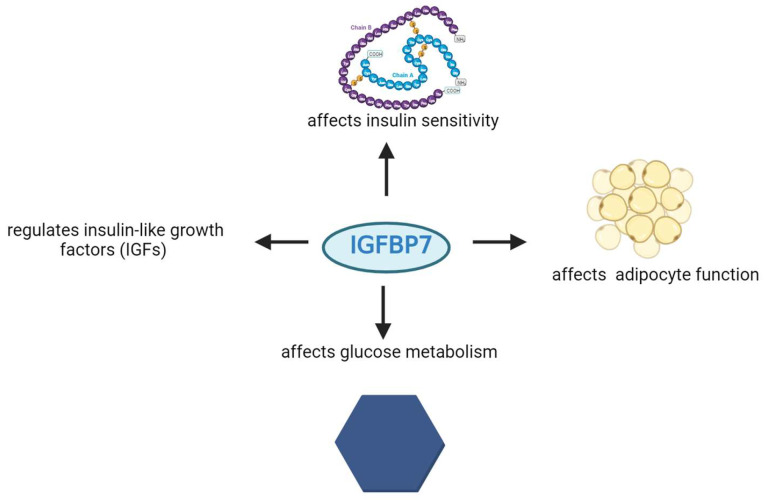
Involvement of IGFBP-7 protein in specific metabolic processes.

**Table 1 biomedicines-12-00159-t001:** Prediction and risk assessment role of GLP-1, GIP, MCP-1 and IGFBP7 in diabetes or obesity. GLP-1—Glucagon-like peptide-1; GIP—Glucose-dependent insulinotropic polypeptide; MCP-1—monocyte chemoattractant protein 1; IGFBP7—Insulin-like growth factor-binding protein 7.

Biomarker	Prediction Role	Risk Assessment Role
**GLP-1**	Reduced basal GLP-1 levels or impaired GLP-1 response postprandially may indicate a diminished incretin effect, contributing to type 2 diabetes and obesity.	Elevated fasting serum GLP-1 concentrations could suggest compensatory mechanisms in response to insulin resistance. Monitoring GLP-1 response to glucose challenges provides insights into beta cell function and its role in obesity-associated insulin resistance.
**GIP**	Altered GIP levels, especially post-glucose ingestion, may indicate disruptions in insulin secretion and potential risks for type 2 diabetes and obesity.	Increased fasting serum GIP levels, particularly after glucose ingestion, could signal impaired glucose regulation and contribute to the development of both diabetes and obesity.
**MCP-1**	Increased levels of MCP-1 in the blood may be linked to inflammatory processes that are connected to obesity, type 2 diabetes, and insulin resistance.	Monitoring MCP-1 levels, especially in the context of periodontal disease, can provide insights into the inflammatory component of diabetes and obesity-related inflammation.
**IGFBP7**	Changes in IGFBP-7 levels may be associated with insulin resistance, metabolic syndrome, early kidney injury, and obesity-related metabolic dysfunction.	Increased concentrations of IGFBP-7, especially in urine, may indicate a higher risk of metabolic syndrome, insulin resistance, potential kidney injury, and its association with obesity-related metabolic complications.

**Table 2 biomedicines-12-00159-t002:** Biomarkers’ commonality in T2D and obesity, their concentration changes across stages, and quantitative differences. GLP-1—Glucagon-like peptide-1; GIP—Glucose-dependent insulinotropic polypeptide; MCP-1—monocyte chemoattractant protein 1; IGFBP7—Insulin-like growth factor-binding protein 7.

Biomarker	Commonality	Changes across Stages	Quantitative Differences
GLP-1	GLP-1 is implicated in both diabetes and obesity. In early stages, there might be compensatory increases, while in later stages, reduced levels could contribute to impaired insulin response.	Early stages may show elevated GLP-1 concentrations as a response to insulin resistance. In later stages, a decline might occur, impacting glycaemic control and potentially contributing to the progression of diabetes and obesity.	Quantitative levels can differ, with lower levels in severe illness states and larger concentrations in early compensatory stages.
GIP	GIP is linked to both diabetes and obesity. Elevated fasting GIP levels may be observed, especially after glucose ingestion.	Early stages may exhibit increased GIP as a compensatory response. In later stages, this elevation might contribute to insulin resistance, diabetes, and obesity.	In early phases, higher concentrations are observed, and in later stages, there may be changes in the quantitative levels.
MCP-1	MCP-1 is associated with inflammation in diabetes and obesity. Concentrations may increase with the progression of both conditions.	Early stages may show moderate MCP-1 elevations, while in later stages, especially with periodontal disease, a significant increase may occur, exacerbating inflammation in diabetes and obesity.	Quantitative levels may vary, with a more pronounced increase in later stages, especially in the presence of periodontal disease.
IGFBP7	IGFBP-7 is linked to insulin resistance, metabolic syndrome, and obesity-related complications.	Early stages may exhibit changes in IGFBP-7 associated with insulin resistance. In later stages, especially with kidney injury, concentrations may rise, indicating a higher risk of metabolic complications in diabetes and obesity.	Quantitative levels may differ across stages, with potential increases in later stages, particularly in the context of kidney injury and metabolic syndrome.

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
