# Peer review of "The Role of GLP-1, GIP, MCP-1 and IGFBP-7 Biomarkers in the Development of Metabolic Disorders: A Review and Predictive Analysis in the Context of Diabetes and Obesity"

_biomedicines, 2024, doi:10.3390/biomedicines12010159_

Round 1

Reviewer 1 Report

Comments and Suggestions for Authors

Author Response

Dear Reviewer1,

Thank you for your suggestions.

The title was improved now according to your first suggestion. The discussion in line 40-42 was re-written as you suggested. Line 57 and 71 have references added. The drawback of methods for T2D was established. Thank you for your remark regarding the biomarkers order, however the purpose of the study was to establish markers specific both for obesity and diabetes mellitus type 2. Therefore, we cannot resign from either of them, as it would contradict the aim of the study. Regarding the details required for the Table 1, we decided to remove it, as we are not able to provide all requested details.

The layout of the discussion was improved. All the abbreviations in the manuscript are described only when they are mentioned for the first time.

To answer your question regarding ‘validating biomarkers’ please see the explanation below: Validating biomarkers involves a rigorous process of assessing and confirming the reliability, accuracy, and clinical relevance of certain indicators or measurements that can be used to reflect specific pathophysiological processes in a disease. Biomarkers can be various substances or characteristics that are objectively measured and evaluated as indicators of normal biological processes, pathogenic processes, or responses to therapeutic interventions.

Clinical validity evaluates the association between the biomarker and the clinical outcome or condition it is intended to represent. This step involves studying the biomarker in a diverse population to determine how well it discriminates between different disease states and healthy individuals. Biomarkers should be specific to the disease or condition of interest, and they should also be sensitive enough to detect relevant changes. Specificity ensures that the biomarker is not falsely elevated in the absence of the condition, while sensitivity ensures it can accurately detect the presence of the condition.

Reviewer 2 Report

Comments and Suggestions for Authors

I’ve read with attention the review by JÄ™drysik et al. that is interesting, well-organized, overall well-written and update.

Abstract: it is long and unbalanced towards the intro section. Since it is not a systematic review, probably the abstract should not divided in chapters.

Intro: it is long and should not divided in subchapters

Main text: it should be divided in numbered subchapters

References: They are not reported in the Journal required style

Comments on the Quality of English Language

The text should be improved. Some sentence are really long and hard to be understood.

Author Response

Dear Reviewer 2,

Thank you for your remarks.

The abstract was shortened and balanced towards the intro section as suggested. The text was consolidated. Intro section was shortened and balanced without the subchapters. Additionally, the main text was divided into subchapters as suggested. The references style was corrected, we apologize for having overlooked the mistakes before.

Reviewer 3 Report

Comments and Suggestions for Authors

1. This manuscript mainly reviews the role of GLP-1, GIP, MCP-1 and IGFBP-7 biomarkers in the 2 development of metabolic disorders , but there is a lack of accurate data on the prediction and risk assessment of these biomarkers in diabetes or obesity. Please use table to present the related results.

2. Are these biomarkers (GLP-1, GIP, MCP-1 and IGFBP-7) common to diabetes and obesity? How do these markers change in the early, middle and late stages of these two diseases? Are there differences in the quantitative level of the same marker at different stages of the disease?

3. In the introduction, the authors should at least mention make clear the relationship between diabetes and obesity, the similarities and differences, as well as the similarities and differences in the markers, which are best shown in a visualized Figure.

4. There are also multiple inconsistencies in the indentation and the line spacing formats such as line 34, 35-39,74,360,386,403 et al.

5. The font in the Figure 1 (such "Results in normal insulin secretion" ) is too small to see clearly.

6. There are too many typo, grammar, punctuation, spacing, and format errors. References: the format of the references is not homogeneous. The Spaces before parentheses are inconsistent in the references cited such as 369 &373. 

Comments on the Quality of English Language

Minor editing of English language required.

Author Response

Dear Reviewer 3,

Thank you for your suggestions.

Regarding point 1 and 2, the accurate data on the prediction and risk assessment of GLP-1, GIP, MCP-1, and IGFBP-7 biomarkers in diabetes or obesity but also its change across stages and differences in the quantitative level of the same marker at different stages of the disease, have been incorporated into a table 1 and 2 for clarity.

Additionally, the manuscript now includes an improved introduction, that elucidates the relationship between diabetes and obesity, emphasizing both similarities and differences, along with the corresponding markers. The inconsistencies in indentation and line spacing formats, as well as font size issues in Figure 1, have been rectified for better readability. Thank you for noticing it.

Furthermore, we have meticulously corrected multiple typo, grammar, punctuation, spacing, and format errors throughout the manuscript. We apologize for having overlooked the mistakes before.

 The references have been homogenized in format, and the spacing before parentheses has been made consistent, addressing concerns raised in references 369 and 373.

Round 2

Reviewer 1 Report

Comments and Suggestions for Authors

I am satisfied with the progress made.

Reviewer 3 Report

Comments and Suggestions for Authors

Dear Authors,

In the revised version of the manuscript "The role of GLP-1, GIP, MCP-1 and IGFBP-7 biomarkers in the development of metabolic disorders: a review and predictive analysis in the context of diabetes and obesity", the authors have incorporated suggestions and answered questions.

Best regards